# Serving Two Masters: Effect of *Escherichia coli* Dual Resistance on Antibiotic Susceptibility

**DOI:** 10.3390/antibiotics12030603

**Published:** 2023-03-17

**Authors:** Olusola Jeje, Akamu J. Ewunkem, Liesl K. Jeffers-Francis, Joseph L. Graves

**Affiliations:** 1Biology Department, North Carolina Agricultural and Technical State University, 1601 E Market Street, Greensboro, NC 27411, USA; 2Department of Biological Sciences, Winston Salem State University, 601 S Martin Luther King Jr Drive, Winston Salem, NC 27110, USA

**Keywords:** *Escherichia coli*, T7 bacteriophage, antibiotic, iron (III), experimental evolution, antimicrobial-resistance

## Abstract

The prevalence of multidrug-resistant bacteria and their increased pathogenicity has led to a growing interest in metallic antimicrobial materials and bacteriophages as potential alternatives to conventional antibiotics. This study examines how resistance to excess iron (III) influences the evolution of bacteriophage resistance in the bacterium *Escherichia coli*. We utilized experimental evolution in *E. coli* to test the effect of the evolution of phage T7 resistance on populations resistant to excess iron (III) and populations without excess iron resistance. Phage resistance evolved rapidly in both groups. Dual-resistant (iron (III)/phage) populations were compared to their controls (excess iron (III)-resistant, phage-resistant, no resistance to either) for their performance against each stressor, excess iron (III) and phage; and correlated resistances to excess iron (II), gallium (III), silver (I) and conventional antibiotics. Excess iron (III)/phage-resistant populations demonstrated superior 24 h growth compared to all other populations when exposed to increasing concentrations of iron (II, III), gallium (III), ampicillin, and tetracycline. No differences in 24 h growth were shown between excess iron (III)/phage-resistant and excess iron (III)-resistant populations in chloramphenicol, sulfonamide, and silver (I). The genomic analysis identified selective sweeps in the iron (III) resistant (*rpoB, rpoC, yegB, yeaG*), phage-resistant (*clpX* →/→ lon, *uvaB, yeaG, fliR*, gatT, *ypjF, waaC, rpoC, pgi, and yjbH*) and iron (III)/phage resistant populations (*rcsA, hldE, rpoB*, and *waaC*). *E. coli* selected for resistance to both excess iron (III) and T7 phage showed some evidence of a synergistic effect on various components of fitness. Dual selection resulted in correlated resistances to ionic metals {iron (II), gallium (III), and silver (I)} and several conventional antibiotics. There is a likelihood that this sort of combination antimicrobial treatment may result in bacterial variants with multiple resistances.

## 1. Introduction

Antimicrobial resistance (AMR) is a major public health challenge [1,2]. This trend portends danger in healthcare delivery as a United Nations interagency group report on antibiotic resistance predicted a likelihood of 10 million deaths per year worldwide by 2050 [3]. This has led to calls for developing alternatives to antibiotic therapy, such as ionic and nanoparticle metals and bacteriophages [4,5]. Bacteriophages are bacteria-specific viruses that lyse and kill infected bacteria. They are currently used in agriculture and husbandry to kill bacteria found in food products, including fruits, vegetables, fish, frozen foods, and cheese, as well as to control the colonization of bacteria in poultry and pigs [6,7,8,9,10]. Ionic and nanoparticle metals have also been proposed as potential antimicrobials [4,11]. Both ionic and nanoparticle iron have been shown to have potent anti-bacterial effects [12]. As iron metabolism is fundamental to bacterial homeostasis, excess iron has been shown to be effective across bacterial phylogeny [13]. Excess iron under aerobic conditions has been shown to induce oxidative damage. The interaction of iron (II) and (III) with hydrogen peroxide and superoxides generate highly reactive hydroxyl radicals, which subsequently lead to cell damage and death [14]. Bacteria cells have genes to defend themselves in response to the formation of oxygen free radicals. OxyR is a protein that responds to the presence of hydrogen peroxide, SoxS, and SoxR responds to redox-active compounds, while RpoS responds to general oxidative stress [15,16,17]. Moreover, several experiments have shown that bacteria can rapidly evolve resistance to both ionic and nanoparticle metals [16,18]. These studies also demonstrated that the evolution of ionic and nanoparticle metal resistance resulted in correlated resistance to antibiotics [16,18]. For instance, Graves et al. and Thomas et al. reported that *E. coli* could rapidly evolve resistance to gallium (III) nitrate and iron (II) sulfate, which subsequently demonstrated correlated resistance to conventional antibiotics [18,19]. Bacteria cultured as biofilms may also evolve heritable variation for resistance to antibiotics de novo [20,21]. It has been suggested that this variation in antibiotic resistance may arise with time in biofilms, even in the absence of antibiotic selection [21]. This could be driven by an accumulation of neutral variation or the selection of phenotypes correlated with antibiotic resistance [22]. A study on *E*. *coli* RP437 also suggested a level of dynamism in antibiotic–biofilm interaction [23]. *E*. *coli* RP437 was more susceptible to antibiotics during its early-stage biofilm formation than in later stages [23]. However, the strain of *E. coli* in the present study was not grown as a biofilm.

Bacteria have demonstrated that they rapidly evolve resistance against single toxins [16,18,19]. However, the evolution of resistance to a specific toxin may come at a cost to other aspects of fitness [24]. Thus, if the bacterium becomes better at resisting one toxin, it may simultaneously become worse at resisting a second. The impact of an organism’s prior evolutionary history upon its potential to evolve new phenotypes is called fitness epistasis [25]. For example, bacteria may become resistant to antibiotics via efflux pumps. However, these efflux pumps can often serve as receptors for bacteriophages [5,11]. Thus, their past evolution of antibiotic resistance makes them vulnerable to a variety of phages [26]. One recent study tested this by using engineered bacteriophages successfully administered to treat a patient with a disseminated drug-resistant *Mycobacterium abscessus* [26].

In this study, we utilize experimental evolution to first generate bacterial resistance to excess iron (III), followed by the evolution of phage T7 resistance to test the fitness epistasis in *E. coli*. Graves et al. reported that excess iron (III) resistance occurred in *E. coli* K-12 MG 1655 associated with a mutation in outer membrane protein C (*ompC*) [11]. Thus, excess iron resistance may also involve variation in outer membrane proteins, which in turn influence how they evolve resistance to phages. Furthermore, resistance to bacteriophages often involves mutations in the proteins phages use to enter bacterial cells. The outer membrane receptor, *OMR,* unto which coliphage T7 binds to initiate infection, is the inner core (IC) region of LPS [27,28]. Thus, we wanted to test the effect of sequential selection utilizing excess iron (III) and phage T7 on correlated fitness components in *E. coli.*

## 2. Materials and Methods

### 2.1. Strains and Growth Conditions

*Escherichia coli* K-12 MG1655 (ATCC #47076) was used for this study due to the rarity of known metal or antibiotic-resistant loci in this bacterium [18]. There are no plasmids in this strain, and the circular chromosome comprises 4,641,652 nucleotides (GenBank: NC_000913.3) [29]. Lytic *Escherichia coli T7* bacteriophage was provided by Dr. Christina Burch (UNC-Chapel Hill). The phage titer was determined (3.4 × 10^6^ pfu) and subsequently stored temporarily at−20 °C for short-term usage. For long-term storage, aliquots of filtered T7 bacteriophage were mixed with 50% (*v*/*v*) glycerol in ratio of 50:50 and stored at −80 °C. Stock T7 phage was subsequently diluted using 0.86% (*w*/*v*) NaCl, also called phage juice.

*E. coli* K-12 MG1655 was grown in Davis Minimum Broth (DMB, Difco™, Sparks, MD, USA) fortified with 10% (*w*/*v*) dextrose (Dextrose, Fisher Scientific, Fair Lawn, NJ, USA) as the only source of carbon and enriched with 0.1% (*w*/*v*) Thiamine hydrochloride in 10 mL of total culture volume. Cultures were maintained in 50 mL Erlenmeyer flasks at 37 °C with shaking at 150 rpm for 24 h. The stock culture was propagated by daily transfers of 0.1 mL of each culture into 9.9 mL of DMB for 7 days of regrowth before selection for iron (III) resistance began. The controls were set up by transferring five different 0.1 mL samples and adding them to 9.9 mL of DMB broth. These cultures were grown for 24 h, representing 6.5 generations of population growth from ∼10^6^ cells per mL at hour zero to 10^8^ cells per mL at 24 h.

### 2.2. Evolution Experiment

Ten flasks of *E. coli* K-12 MG1655 were exposed to 1.75 mg/mL iron (III) sulfate solution. Another set of five flasks was not exposed to iron (III) treatments to serve as control. The transfer was done daily while iron (III) MIC was carried out at a 7-day interval, and growth curves were plotted to determine phenotypic differences among populations. Evidence of excess iron (III) resistance was generally observable on or before day 21 of iron (III) treatments. This is measured by superior cell density in growth curves during iron (III) MIC. When evidence of iron (III) resistance was observed, the 10 iron (III)-resistant flasks were split into 20. Ten flasks were further selected for T7 bacteriophage resistance, while the second set of 10 flasks continued to be exposed to iron (III) only. The 5 control flasks were split into 10; 5 flasks were selected for T7 phage and continued to be exposed to the same, while the last 5 flasks served as control. Overnight culture of the ancestral line was prepared and used during assays. Bacterial populations were grown for a minimum of 35 days.

### 2.3. Bacteriophage Resistance Assay

To develop phage-resistant bacterial populations, 400 µL of T7 bacteriophage from stock (3.4 × 10^6^ pfu) was pipetted on DMB agar plate, 100 µL of iron (III)-resistant, overnight bacterial culture was added to the center of the phage droplet on the DMB agar plate. The pool was mixed with a spreader and allowed to air-dry. The same was done on the 5 control groups (not selected for iron (III) or bacteriophage). Plates were covered and stored in the incubator at 37 °C for 24 h. Bacteriophage-resistant colony was picked from each plate to establish dual iron (III)/phage-resistant and phage-only-resistant populations. The colonies were cultured in DMB medium in 50 mL Erlenmeyer flasks. The new colonies were grown for 48 h in DMB medium before further treatment with iron (III) and bacteriophage. Bacteriophage-resistant tests were performed by making 3 straight, horizontal streaks of bacteria on a fresh DMB plate using a cotton swab; one straight, vertical streak of undiluted T7 bacteriophage runs through the middle of the bacteria streaks. Plates were covered and stored at 37 °C for 24 h. Plates were observed after 24 h for evidence of phage resistance. Bacteriophage-resistant populations developed were exposed to 1:100 dilution of stock T7 bacteriophage.

### 2.4. Phenotypic Assays: 24 h Minimum Inhibitory Concentration Growth

Iron (III)-selected, iron (III)/phage-selected, phage-selected, and control populations were given fresh media daily while resistance to iron (III), iron (II), silver (I), and gallium (III) was measured using 24 h minimum inhibitory concentration (MIC) growth assays [30]. MIC is often user-defined; however, for the present study, we used “lowest concentration of a particular substance needed to inhibit the growth of a certain population of bacteria” [18,30]. Our MIC is thus the concentration of tested substances that inhibited any visible growth of the organism over 24 h. MICs were determined via serial dilution. Antibiotic resistance was measured in all populations of iron (III)-resistant, iron (III)/phage-resistant, phage-resistant, control, and ancestral populations using 24 h growth in increasing concentration assays [31]. Concentrations of test solutions (ampicillin, sulfonamide, rifampicin, tetracycline, and chloramphenicol) were 0.00 mg/mL, 6.0 mg/mL, 12 mg/mL, 25 mg/mL, 50 mg/mL, 75 mg/mL, 100 mg/mL, 175 mg/mL, 250 mg/mL, and 500 mg/mL. Overnight cultures of each sample were used with ten-fold dilution of the five drugs. Growth curves were used to assess population density between tests and control and subsequently as a measure of fitness of bacteria populations in tested substances [18]. Bacterial growth in DMB was assessed by measuring turbidity at 620 nm for hours 0, 4, 8, 12, and 24, using a 98-well plate Synergic Mx spectrophotometer (Biotek, Henrico, VA, USA) using clear polyester 98-well plates.

### 2.5. T7 Bacteriophage Resistance Assay

Plaque assay was carried out to determine the susceptibility of the five populations to lytic phage attack. LB plates were used because they provided the necessary contrast to visualize plaques formed using DMB soft agar. Overnight cultures of the populations were used. Five milliliters of the melted soft agar were added to 400 uL of overnight culture of each bacterial population. Stock T7 phage (3.4 × 10^6^ pfu) was serially diluted in 0.86% (*w*/*v*) NaCl, and 100 µL of 1:50 phage titer was pipetted into the soft agar-bacteria mixture. Tubes were rubbed for a homogenous mix of bacteria, soft agar, and phage. The mixture was poured on the LB plate, rocked gently to spread, allowed to solidify, covered, and incubated at 37 °C. Plaques were visible and estimated after 3 h. Counts were recorded and analyzed for the 12 replicates per population.

### 2.6. Dual Resistance Assay: Excess Iron (III) and Escherichia Phage T7

Ancestral population was cultured overnight prior to the assay as a control. In a 50 mL Erlenmeyer culture flask, populations were treated with 1.40 mg/mL (1400 µL) of iron (III) and 100 µL of stock T7 phage and incubated for 24 h. The 1.40 mg/mL (1400 µL) iron (III) concentration was used for this assay because all populations were susceptible to excess iron (III) at that concentration threshold. DMB plates were pre-warmed at 37 °C in 10 replicates per population. Serial dilutions of each population were transferred to DMB plates and incubated overnight. Ten microliters of diluted bacteria culture were pipetted into the DMB plate. The inoculum was spread evenly on the plate with the aid of a cell spreader. Plates were covered and incubated at 37 °C for 24 h. Total number of bacteria colonies was counted.

### 2.7. Genomic Analysis

DNA was extracted from each population after 35 days of culture in excess iron (III) using the EZNA Bacterial DNA extraction kit (Omega Bio-tek^®^) according to the manufacturer’s instructions; DNA concentrations were normalized using the QuantiFluor^®^ dsDNA system [18]. Genomic libraries were prepared using the Illumina Nextera XT kit, and samples were sequenced using the Illumina MiSeq sequencing platform. The depth of coverage of the sequencing runs ranged from ∼20× to ∼80×, with most exceeding 40× coverage. The SRA accession number for sequencing data from this study is PRJNA803149 (iron (III)/phage-resistant, iron (III)-resistant, phage-resistant, and control).

Genomic variants were called via the breseq 33 pipeline per our previous studies [16]. The breseq pipeline uses three types of evidence to predict mutations, read alignments, missing coverage, and new junctions, and any reads that indicate a difference between the sample and the reference genome that cannot be resolved to describe precise genetic changes are listed as ‘unassigned’ [32]. The algorithm computes frequency by the number of reads that contain the de-novo mutation. Ten replicates of iron (III)-resistant populations were sequenced, codenamed ‘Fe1, Fe2…Fe10’. Four replicates of bacteriophage-resistant populations were successfully sequenced, codenamed ‘Ph2, Ph3…Ph5’. Five replicates of iron (III)/phage-resistant populations were codenamed ‘FPh1, FPh2, FPh4….6’, and five controls were sequenced and codenamed ‘Ctrl1, Ctrl2…Ctrl5’.

### 2.8. Statistics

Growth and cell density were measured using growth curves constructed with the GraphPad Prism software (v.8.1, GraphPad Software Inc., La Jolla, CA, USA). The 24 h growth phenotypes in response to increasing concentrations of metals and antibiotics were analyzed using the IBM SPSS general linear model. This software computes a two-way analysis of variance (ANOVA) for the variables: population and concentration. The software computes the F values for the effect of population, concentration, and their interaction on 24 h growth. The analysis of the mean resistance to lytic phage in the plaque assay, as well as to the combination of lytic phage and excess iron, was determined by ANOVA and compared using the Tukey post hoc multiple comparison tests.

## 3. Results

### 3.1. The Effect of Sequential Iron (III), Bacteriophage Selection on Metal Resistance

To determine the effect of sequential selection on metal resistance, we assessed growth in excess iron (III), iron (II), gallium (III), and silver (I). Figure 1a showed that iron (III)/phage-resistant populations showed superior growth compared to iron (III)-resistant, phage-resistant, control, and ancestor populations in increasing concentration of iron (III). Phage-resistant populations showed superior growth in comparison to iron (III)-resistant, ancestral, and control populations in increasing iron (III) concentrations. In increasing concentration of gallium (III), iron (III)-resistant populations showed superior growth compared to all other populations (Figure 1b). While the iron (III)/phage-resistant populations showed superior growth compared to phage-resistant, control, and ancestor populations (Figure 1b). In Figure 1c, with increasing iron (II) concentrations, there was no significant difference in 24 h growth between iron (III)/phage-resistant populations relative to iron (III)-resistant populations. However, iron (III)/phage-resistant populations demonstrated superior growth in comparison to all the other populations. In increasing gallium (III), iron (II), and silver (I), the differences observed in the growth of the phage-resistant and control populations were not significant (Figure 1b–d).

The iron (III)/phage- and iron (III)-selected populations showed superior growth relative to the phage-selected, control, and ancestral strain across concentrations for silver (I) (Figure 1d). Surprisingly, however, at the highest silver concentration (100 mg/L), there is an increase in the growth of the iron (III) and iron (III)/phage populations. The genomic analysis showed no selection in genes (such as cusS and ompR) associated with silver resistance. In our previous studies, the selection of minerals whose primary component is iron or share common chemical properties with iron (e.g., gallium and magnetite) confer minor increases in silver resistance [16,18]. Thus, it is possible that other mechanisms associated with iron (III) resistance, yet to be determined, could account for an increase in silver resistance at the highest concentrations measured.

Table 1 shows the general linear model (GLM) results for phenotypic assays comparing iron (III)/phage-resistant populations to all other populations, including the F-statistics and *p*-values for all phenotypic comparisons. Relative comparison between populations of iron (III)/phage-resistant and control showed interactions between population and concentration variables in increasing concentrations of iron (II), iron (III), gallium (III), and silver (I). The same effect was observed relative to iron (III)/phage and ancestor populations. There was no interaction between concentration and population variables when iron (III)/phage-resistant and iron (III)-resistant populations were compared with increasing metal concentrations (Table 1). Comparison of iron (III) versus phage showed interactions between population and concentration variables in increasing concentrations of iron (II), gallium (III), and silver (I). Additionally, iron (III)-resistant populations showed interactions between control and ancestor populations in increasing iron (III), iron (II), gallium (III), and silver (I) (Table 1). When comparing the phenotypic assays among phage vs. control, the general linear model showed no significant interaction effect in increasing iron (III), iron (II), gallium (III), or silver (I) (Table 1). When comparing the phenotypic assays between phage and ancestor, the general linear model showed a significant interaction effect in increasing iron (III), gallium (III), and silver (I), indicating that the functional response to these metals differed between these populations (Table 1). Table 1 showed that in comparison to control and ancestor, there was only a significant interaction effect in increasing gallium (III).

### 3.2. The Effect of Sequential Iron (III), Bacteriophage Selection on Antibiotic Resistance

To determine the effect of sequential selection on correlated antibiotic resistance, we assessed growth in five traditional antibiotics that target major essential functions of bacteria: ampicillin, tetracycline, rifampicin, sulfonamide, and chloramphenicol (Figure 2). In increasing concentration of ampicillin after 35 days of evolution in excess iron (III) and selection for T7 resistance, iron (III)/phage-resistant > iron (III)-resistant > phage-resistant > controls > ancestors (Figure 2a). With increasing tetracycline concentration (Figure 2b), iron (III)/phage-resistant showed superior growth compared to both iron (III)-resistant and phage-resistant populations. Iron (III)-resistant populations showed no difference compared to phage-resistant populations in increasing tetracycline concentrations. Iron (III)/phage-, iron (III)-, and phage-resistant populations all showed superior growth compared to control and ancestor populations in increasing tetracycline concentration (Figure 2b).

With increasing rifampicin concentration (Figure 2c), iron (III)-resistant populations showed superior growth compared to iron (III)/phage-resistant populations. At the same time, there was no significant difference between iron (III)-resistant compared to phage-resistant and iron (III)/phage-resistant compared to phage-resistant in increasing concentrations of rifampicin. Iron (III)/phage-resistant, iron (III)-resistant, and phage-resistant all showed superior growth compared to control and ancestor populations in increasing rifampicin concentration (Figure 2c). With increasing sulphanilamide concentration (Figure 2d), iron (III)/phage-resistant populations showed no significant difference compared to iron (III)-resistant; however, both iron (III)/phage- and iron (III)-resistant populations showed superior growth compared to phage-selected, control and ancestor populations (Figure 2d). With increasing chloramphenicol concentration (Figure 2e), iron (III)/phage-resistant populations showed no significant difference compared to iron (III)-resistant populations. Both iron (III)/phage- and iron (III)-resistant populations showed superior growth compared to phage-resistant populations. Phage-resistant populations showed no significant difference compared to ancestor and control populations in increasing chloramphenicol. Iron (III)/phage-resistant and iron (III)-resistant both showed superior growth compared to control and ancestor populations in increasing chloramphenicol concentration (Figure 2e).

The two-way ANOVA results (Table 1) showed significant interactions between the population and concentration variables with increasing concentrations of tetracycline, chloramphenicol, sulfonamide, and rifampicin, in relative comparison of iron (III)/phage-resistant population to the control and ancestor populations. In increasing concentrations of ampicillin, there were no significant interactions between the population and concentration variables in a relative comparison of the iron (III)/phage-resistant population to the control, ancestor, and iron (III)-resistant populations. Phenotypic assays comparing phage to control showed significant interaction effects only in increasing tetracycline concentration (Table 1). Additionally, phage-resistant populations showed a significant interaction effect compared to ancestor populations in increasing tetracycline, ampicillin, and rifampicin (Table 1). When comparing control populations to ancestor populations, a significant interaction effect was only shown in increasing chloramphenicol and rifampicin (Table 1).

### 3.3. The Effect of Sequential Selection on Bacteriophage Resistance

A resistance assay was performed to determine the resistance of each selected population to both iron (III) and T7 phage. All populations were susceptible to 1.40 mg/mL (1400 µL in volume) excess iron (III) and thus used for this assay. Ranking of populations according to fitness in excess iron (Table 2) showed that iron (III)/phage-resistant > iron (III)-resistant > phage-resistant > control > ancestor populations. Similarly, the ranking of populations based on T7 bacteriophage resistance showed iron (III)/phage-resistant > phage-resistant > iron (III)-resistant > control > ancestral populations. Figure 3 (and Appendix A) showed the susceptibility of each population to T7 bacteriophage lysis by graphing the number of plaques formed by T7-infected bacterial populations. The graph (Figure 3) showed that the ancestor populations had the highest number of plaques (197.33 ± 6.19, *p* <0.001) followed by the control population (147.17 ± 6.19, *p* < 0.001), and iron (III)-resistant population (82.83 ± 6.19, *p* < 0.001)—no plaques formed on the iron (III)/phage-resistant and phage-resistant populations. The statistical mean difference of plaque formation between populations is shown in Appendix A.

To determine dual resistance in each population, an assay was performed to estimate the concurrent resistance of populations to excess iron (III) and lytic T7 phage attack. Bacterial colonies that demonstrated resistance to excess iron (III) and T7 lysis were counted after 24 h of incubation in excess iron (III) and T7 phage. Iron (III)/phage-resistant, phage-resistant, and iron (III)-resistant populations had 130, 109, and 94 resistant colonies, respectively (Figure 4 and Appendix A). However, the mean comparison between iron (III)/phage-resistant with phage-resistant and iron (III)-resistant populations showed that there was no difference in fitness among the populations (*p* = 0.530, *p* = 0.074). In addition, there was no significant difference observed between iron (III)-resistant and phage-resistant populations (*p* = 0.797). Ancestral and control populations had 34 and 47 resistant colonies, respectively, but the difference between the two populations was not significant (*p* = 0.870). There was a significant difference in resistance between the ancestral population and iron (III)-resistant, phage-resistant, and iron (III)/phage-resistant populations (*p* = 0.001, *p* = 0.000, and *p* = 0.000, respectively). The control population also showed inferior 24 h growth relative to iron (III)-resistant, iron (III)/phage-resistant, and phage-resistant populations (*p* = 0.001, *p* = 0.000, and *p* = 0.000, respectively). The statistical mean difference of bacterial colonies between populations is shown in Appendix A.

To confirm bacteriophage resistance in each selected population, a bacteriophage resistance assay was performed, as described in the methods. Figure 5 showed that the iron (III)-resistant and control populations were not resistant to T7 bacteriophage lysis. However, the phage- and iron (III)/phage-resistant populations were resistant to T7 bacteriophage lysis.

### 3.4. Whole Genome Sequencing

To determine the effect of sequential selection on *E. coli* genomic variations, we sequenced replicates from each selected population. The data was generated by using Breseq computational pipeline in polymorphism mode. Table 3, Table 4 and Table 5 list the genomic variants, positions, and mutations found in each selected population. At 35 days, one iron (III) replicate (Fe4) and two iron (III) replicates (Fe 9, Fe10) displayed a selective sweep (yellow) for *rpoC* and *rpoB*, respectively (Table 3). All the phage-resistant replicates displayed select sweeps (yellow), and two out of four (Ph2, Ph3) displayed significant polymorphism (green) at day 35 (Table 4). Table 5 lists the gene, position, mutation, and gene annotation of selective sweeps (yellow) and significant (green) polymorphisms detected in five replicates of iron (III)/phage-resistant populations. Two of the four selective sweeps, in genes *waaC,* and *rpoB,* were also detected in the phage-resistant and iron (III)-resistant populations, respectively. While two selective sweeps, in genes *rcsA* and *hldE,* were unique to the iron (III)/phage-resistant populations (Table 5) at day 35. Appendix A list the minor polymorphisms detected in iron (III)/phage-, iron (III)-, and phage-resistant populations at day 35, respectively.

## 4. Discussion

In this study, we hypothesized that the evolution of resistance to iron (III) and subsequently to phage T7 could occur in *E. coli*. The evolution of phage T7 resistance in iron-resistant and control populations occurred within 24 h. There was no evidence that having prior resistance to iron (III) retarded the capacity to evolve resistance to T7 phage. Indeed, our phenotypic results showed that iron (III)/phage-resistant populations exhibited superior 24 h growth in excess iron (III) as well as superior phage resistance relative to all other populations, including controls (Figure 3). Furthermore, in this experimental evolution of excess iron resistance, no *ompC* mutations were recovered. Thus, these mutations did not drive resistance to iron (III) despite being derived from the same ancestral strain of *E. coli* as those utilized in our previous study in Graves et al. [11]. This failure to recover *ompC*, as well as some other mutations from the first experiment (*murC*, *cueR, fliP, ptsP, ilvG, fecA*, and intergenic mutation between ilvL/ilvX) apart from *yeaG,* while still showing the same suite of correlated responses (resistance to iron (II), gallium (III), silver (I), chloramphenicol, polymixin B, rifampicin, sulfanilamide, and tetracycline) demonstrated that excess iron (III) resistance could evolve via multiple pathways. In addition, in this experiment, iron (III)/phage-resistant populations showed significantly superior growth in excess iron (III), iron (II), gallium (III), silver (I), ampicillin, tetracycline, chloramphenicol, rifampicin, and sulfanilamide compared to control populations; again, demonstrating the relationship between the evolution of excess iron (III) resistance and this suite of correlated traits.

There was no difference in growth between iron (III)/phage-resistant and iron (III)-resistant populations in increasing excess concentrations of chloramphenicol, sulfanilamide, and silver (I) (Figure 1 and Figure 2). This indicates that acquiring resistance to these antimicrobials was likely the result of excess iron (III) as opposed to phage T7 resistance. It further indicates that there was no antibiotic resistance cost to acquiring phage T7 resistance in these populations. The data indicated that resistance to both iron (III) and phage T7 enhanced the resistance of the population to tested metallic substances and conventional antibiotics (Figure 1 and Figure 2). A similar phenomenon was also reported in the animal production industry, where copper and zinc were added to animal feed for their antimicrobial properties. The metals created a selective pressure resulting in the evolution of resistance to both copper and zinc as well as increased resistance in weaned pigs to tetracycline and sulfanilamide [33,34]. Cross-resistance, co-resistance, or pleiotropy develop when microbes use the same resistance mechanisms to defend against different antimicrobials, such as an efflux pump, or when the genes responsible for resistance are linked closely and are transcribed or transferred together [26,35,36]. Furthermore, while bacteria do not generally retain excess genetic material, it has been shown that bacteria do have the capacity for genomic redundancy that can allow for a phenotype to be produced by alternative means [35]. Such capacity is influenced by epistasis and pleiotropy within gene networks. An example of this was shown in macroevolutionary studies of a color phenotype within the eukaryotic plant family Solanaceae demonstrating the multiple evolutions of red flower color by divergent genomic mechanisms [37].

The genomic results also indicated that the frequency of *rpoB* mutations is highest in iron (III)-resistant populations compared to other populations in this study (Table 3). Mutations in the *rpoB* gene, coding for the β subunit of the bacterial RNA polymerase, have been shown to be massively pleiotropic. They have been linked with rifampicin resistance in many microorganisms, including *E. coli* K-12 [38,39]. Other impacts of *rpoB* mutation are associated with a range of secondary effects on bacterial cells. Secondary effects of *rpoB* mutation are evidenced in transcription, cell fitness, bacterial stress response, and virulence [40,41]. The fitness of the iron (III)-resistant population in excess iron (III) in the present study might have been partly due to the secondary effect of *rpoB* mutations. Mutations in typical stress response genes, including *ycgB* and *yeaG,* were observed and may play a role in iron (III) resistance. Both *ycgB* and *yeaG* are two of ten insertions mapped in nine open reading frames (*yciF, yehY, yhjY, yncC, yjgB, yahO, ygaU, ycgB,* and *yeaG*) of unknown function, which appear to be novel members of the σS or rpoS regulon [40]. Adaptation to sustained nitrogen starvation in E. coli has been reported to be impacted by *yeaG* [41]. Expression of *yeaG* is increased during the stationary phase, acid, and salt stress [42]. The general stress sigma factor σS is strongly induced when *E. coli* cells are exposed to various stress conditions, which include starvation, hyperosmolarity, pH downshift, or nonoptimal high or low temperature [43]. A minor variant *cspC* (f = 0.117), a stress protein member of the *cspA* family, was also detected in the iron (III)-resistant population (Appendix A). In addition, *cspC* belongs to a network of genes that facilitate stress-induced mutagenesis (SIM) in E. coli K-12 [44].

The genomic results also indicated that *waaC* mutations played a role in phage T7 resistance in both iron (III)/phage- (Table 5) and phage-resistant (Table 4) populations. A *waaC* mutant has a defect in the LPS core heptose region, triggering a deep-rough phenotype [45]. Ideally, phage T7 is not expected to attack freshly isolated and smooth phenotypes. Therefore, it is unexpected that the bacteria will resolve to deep-rough phenotype to resist or develop resistance to phage attack. However, two *E. coli* K-12 W3110 *waaC* mutants were reported to be resistant to infection by bacteriophage mEp213 [46]. A *hldE* fixation was detected in the iron (III)/phage population, which has previously been shown to catalyze two reactions in the ADP-L-glycero-β-D-manno-heptose biosynthesis pathway and provides one of the building blocks for the inner core region of lipopolysaccharide (LPS) [47]. The outer membrane receptor (OMR), unto which coliphage T7 binds to initiate infection is the inner core (IC) region of LPS [27,28]. The mutations in genes *waaC* (ADP-heptose:LPS heptosyltransferase 1) and *hldE* impact the inner core region of LPS. ADP-heptose:LPS heptosyltransferase I (*HepI*) is the enzyme responsible for the transfer of the first heptose sugar onto the Kdo2 moiety of the lipopolysaccharide inner core (Table 5) [48]. RNA-binding protein Hfq, an RNA chaperone, was also detected in our phage T7-resistant population. Hfq binds small regulatory RNA (sRNAs) and mRNAs to facilitate mRNA translational regulation in response to envelope stress, among other stressors (Appendix A) [49]. An intergenic mutation between *nudE* ←/→ *yrfF* (ADP-sugar diphosphatase NudE) is another major fixation linked to phage resistance Table 5). An *mrcA nudE yrfF* triple mutant has been reported to exhibit phenotypes that include mucoidy, heat sensitivity, growth defects, and resistance to phage or antibiotic drugs [50]. Other mutations found were *rpoB* and *rpoC*, consistent with selective sweeps, and major variants were also found in the phage-resistant population.

The principal regulator of general stress response in the *E. coli rpoS* subunit of RNA polymerase was also detected in the phage-resistant population (Table 4) [42]. This gene is not readily found or completely absent in rapidly growing cells [42]. However, *σS* is repeatedly induced during entry into the stationary phase. It can also be induced in many other stress conditions, and it is essential for the expression of multiple stress resistances [42], which is required for phage- and iron (III)/phage-resistant populations to survive. Thus, *rpoS* is usually considered a second vegetative sigma factor with a major impact on stress tolerance and, beyond that, on the entire cell physiology under nonoptimal growth conditions [42]. Other mutations observed in the phage-resistant population might have been induced due to growth in a minimal medium, which induces additional stress on the populations. They include *clpX,* an ATP-dependent molecular chaperone that serves as a substrate-specifying adaptor for the *clpP* serine protease in the *ClpXP* and *ClpAXP* protease complexes. The *clpX* gene is a member of the AAA+ (ATPases associated with diverse cellular activities) family of ATPases [51] (Table 4). *ClpX* is required for adaptation to and extended viability in the stationary phase, as well as growth in SDS [52]. Further evidence that mutations *clpX* may be common in adaptation to minimal medium is that these mutations were routinely observed in the Lenski LTEE (*E. coli* B cells grown in the same medium used in this study [35]). In addition, a mutation in *uxaB*, Altronate oxidoreductase, the second enzyme of the galacturonate catabolism pathway, catalyzing the reversible NADH-dependent reduction of D-tagaturonate to D-altronate was also detected in the phage-resistant population [53]. *UxaB* is sensitive to catabolite repression; expression is suppressed in the presence of preferred carbon sources [54].

A mutation in *lapC,* formerly *yejM* (lipopolysaccharide signal transducer LapC), is a major variant also found in phage-resistant populations. It encodes an essential inner membrane protein implicated in lipopolysaccharide (LPS) homeostasis. A *lapC* allele (lapC1163) (expressing a C-terminally truncated form of *lapC*) has been reported to increase outer membrane permeability. Mutations in phage-resistant and iron (III)/phage-resistant populations majorly target the LPS, essential to Escherichia T7 phage infection. *Pgi* expression has been reported to be induced by oxidative stress as a pgi deletion mutant is hypersensitive to oxidative stress induced by paraquat [55]. Pgi belongs to a network of genes that facilitate stress-induced mutagenesis (SIM) in *E. coli* K-12 [44]. The *yjbEFGH* operon produces extracellular polysaccharide [45,56]. *YjbH* may be a lipoprotein and/or an outer membrane porin, and the expression has been reported to be higher in rpoS mutants, which was detected in all samples of phage-selected populations both as selective sweep and major variant [57]. A *rcsA* is a positive DNA-binding transcriptional regulator that belongs to the *LuxR*/*UhpA* family of transcriptional regulators. Its detection in the phage-resistant population might be responsible for some of the success of this population in antibiotics. Members show moderately increased resistance to kanamycin and 2-fold increased β-lactam resistance via increased *ampC* expression [58].

## 5. Conclusions

These studies have shown that *E. coli* K-12 MG1655 can rapidly evolve in succession resistance to excess iron (III) followed by that to bacteriophage. In addition, this selection regime subsequently enhanced the resistance of the bacteria to conventional antibiotics and metallic antimicrobial materials. These previous substantiated studies in this bacterium show the pleiotropic impacts of genomic variants associated with excess iron resistance. We did not, however, show the repetition of specific mutations in the *ompC* gene encoding an outer membrane protein C. Thus, this experiment showed no evidence of dramatic fitness epistasis or a possible tradeoff between excess iron (III) and T7 phage resistance. Furthermore, as the selection in this design was sequential, we do not know whether trade-offs between excess iron (III) and phage T7 might not evolve in a simultaneous selection design. In summary, more studies are needed to determine if a combination therapy of metal and phage will be effective in preventing the evolution of resistance.

## Figures and Tables

**Figure 1 antibiotics-12-00603-f001:**
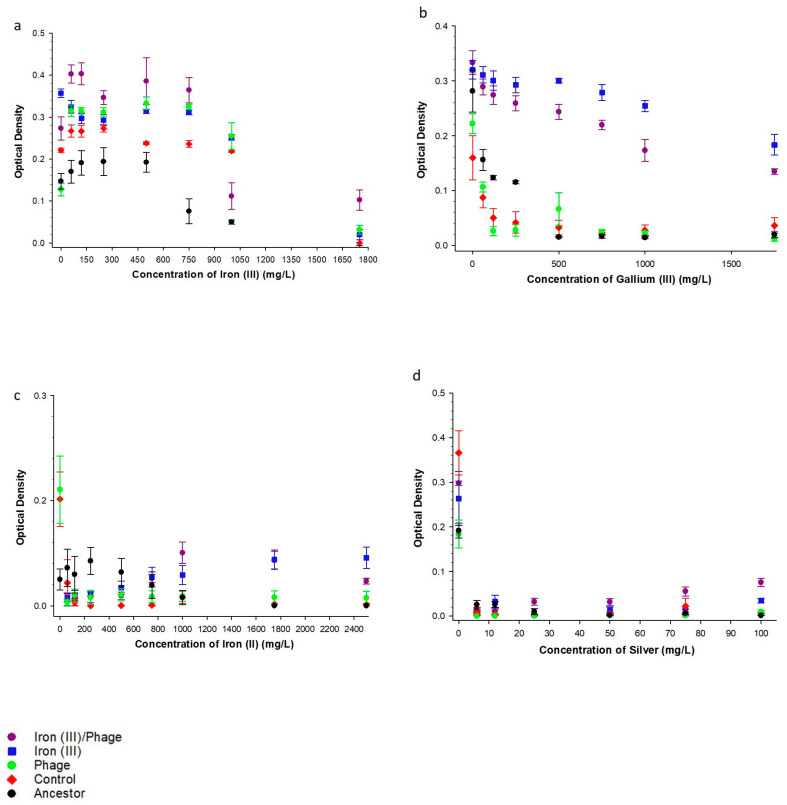
24 h growth fitness in iron, gallium (III), and silver (I). (**a**) The mean and SE of 24 h growth for populations in increasing concentrations of iron (III) to 1750 mg/L. (**b**) The mean and SE of 24 h growth for populations in increasing concentrations of gallium (III) to 1750 mg/L, (**c**) The mean and SE of 24 h growth for populations in increasing concentrations of iron (II) to 2500 mg/L, (**d**) The mean and SE of 24 h growth for populations in increasing concentrations of silver (I) to 100 mg/L. All growths were measured after 35 days of evolution in excess iron (III) and Escherichia phage T7.

**Figure 2 antibiotics-12-00603-f002:**
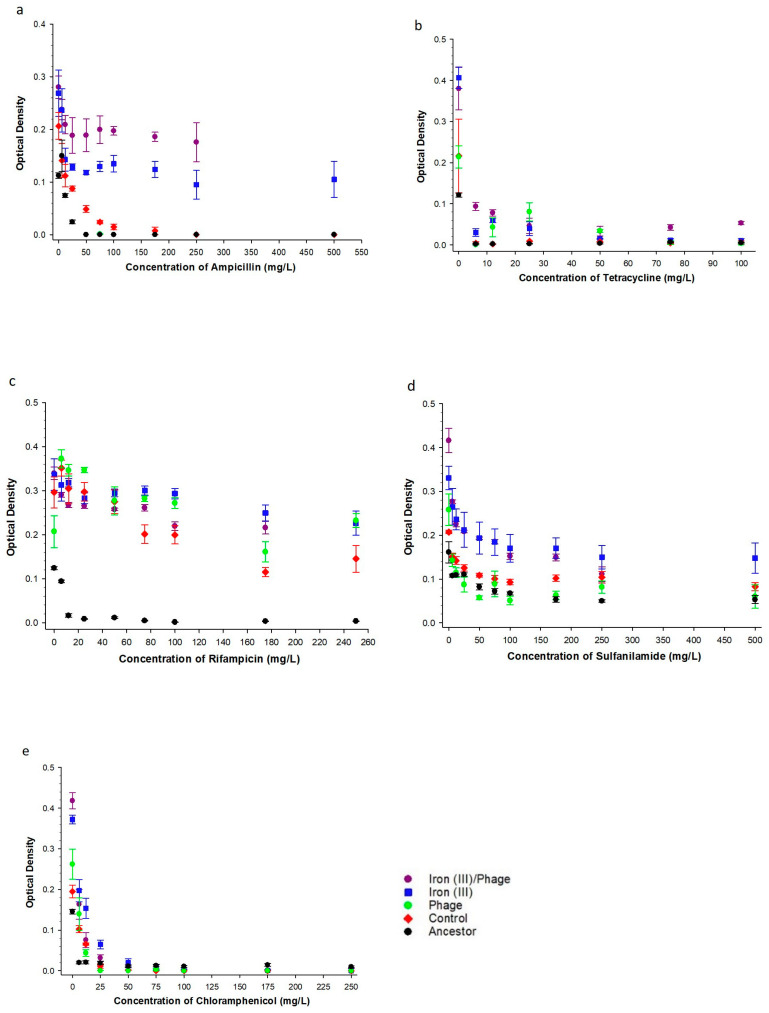
24 h growth fitness in antibiotics. The mean and SE of 24 h growth for populations in increasing concentrations of (**a**) ampicillin to 500 mg/L, (**b**) tetracycline to 100 mg/L, (**c**) rifampicin to 250 mg/L, (**d**) sulphanilamide to 500 mg/L, (**e**) chloramphenicol to 250 mg/L. Growths were measured after 35 days of evolution in excess iron (III) and phage T7.

**Figure 3 antibiotics-12-00603-f003:**
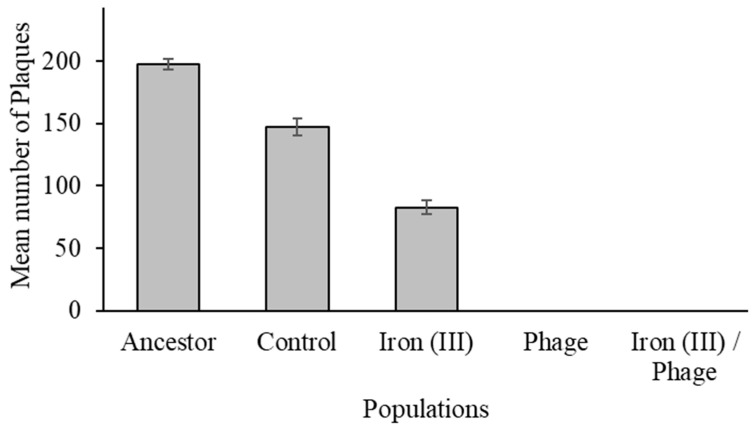
Resistance of populations to Escherichia phage T7. Mean number of plaques represents an average of 12 replicates and SE per population. Phage-resistant and iron (III)/phage-resistant populations were not susceptible to T7 bacteriophage attack. Ancestral population was least resistant to lytic phage attack, followed by the control and iron (III)-resistant populations. Differences in resistance to T7 bacteriophage were significant at 95% confidence interval among populations.

**Figure 4 antibiotics-12-00603-f004:**
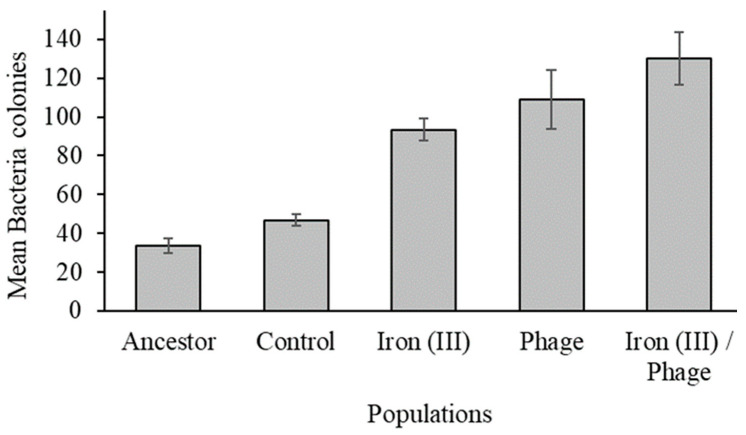
Dual resistance of populations to bacteriophage T7 and iron (III). Mean number of resistant colonies represents an average of 10 replicates and SE per population. Iron (III)/phage-resistant, phage-resistant, and iron (III)-resistant populations, in that order, were better fitted in excess iron (III) and T7 bacteriophage simultaneously. Ancestral population shows least resistance, followed by control. Differences in dual resistance to T7 bacteriophage and iron (III) were significant at 95% confidence interval among populations.

**Figure 5 antibiotics-12-00603-f005:**
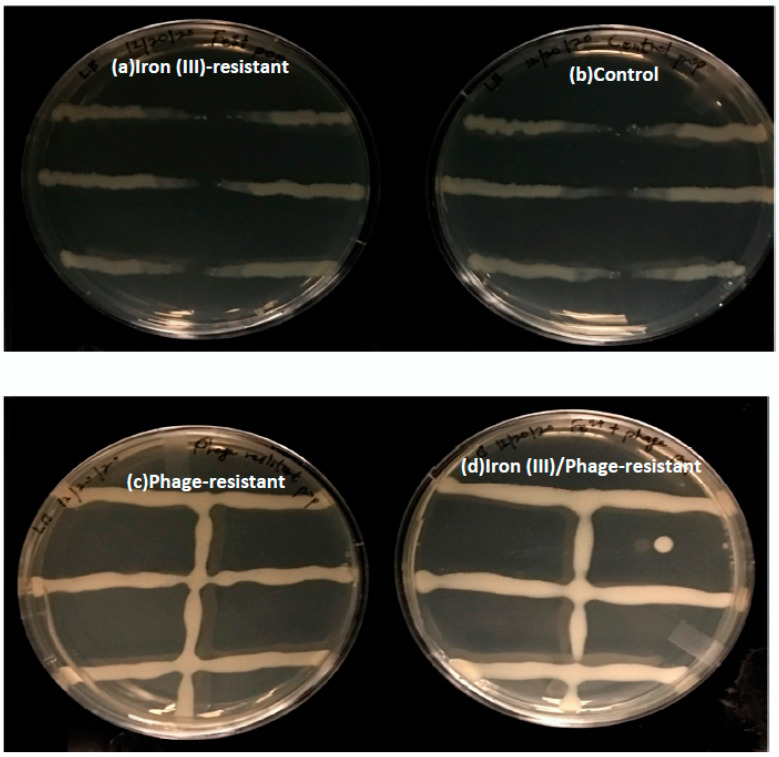
T7 Bacteriophage Resistance test. Representative plates showing bacterial growth in the presence of T7 bacteriophage for (**a**) Iron (III)-resistant *E. coli* population, (**b**) Control population, (**c**) Phage-resistant population, and (**d**) Iron (III)/phage-resistant population. Plates are representative samples of 10 replicates. Absence of a vertical bacteria growth line on plates (**a**,**b**) was an indication that these populations were not resistant to T7 bacteriophage, while the presence of a vertical bacteria growth line on plates (**c**,**d**) was an indication that these two populations were resistant to T7 bacteriophage.

**Table 1 antibiotics-12-00603-t001:** (**a**) Two-way ANOVA computed via the general linear model (GLM) results are shown for phenotypic assays, iron (III)/phage-resistant population compared to all other populations. The general linear model is an analysis of variance to determine the effect of the population (iron (III)/phage), the concentration (range tested for each substance), and their interaction. In addition, the GLM computes an F-statistic and the probability of achieving that F-statistic by chance (*p*-value). (**b**) Two-Way ANOVA computed via general linear model results are shown for phenotypic assays, iron (III) vs. phage vs. control vs. ancestor. (**c**) Two-Way ANOVA computed via general linear model results are shown for phenotypic assays, phage vs. control vs. ancestor. (**d**) Two-Way ANOVA computed via general linear model results are shown for phenotypic assays, control versus ancestor.

**(a)**				
**Substance**	Range Tested	Population	Concentration	Interaction
**Iron (III)/Phage > Controls**				
**Iron (III)**	100–1750 mg/L	F = 10.88, *p* = 0.002	F = 133.21, *p* = <0.001	F = 4.67, *p* = <0.001
**Iron (II)**	100–5000 mg/L	F = 40.58, *p* = <0.001	F = 1.84, *p* = 0.090	F = 3.34, *p* = 0.004
Gallium (III)	100–1750 mg/L	F = 653.99, *p* = <0.001	F = 27.88, *p* = <0.001	F = 15.72, *p* = <0.001
Silver (I)	5–100 mg/L	F = 157.74, *p* = <0.001	F = 42.87, *p* = <0.001	F = 43.99, *p* = <0.001
**Ampicillin**	20–500 mg/L	F = 82.44, *p* = <0.001	F = 9.62, *p* = <0.001	F = 1.31, *p* = 0.258
**Tetracycline**	5–100 mg/L	F = 60.13, *p* = <0.001	F = 33.82, *p* = <0.001	F = 21.11, *p* = <0.001
**Chloramphenicol**	5–250 mg/L	F = 39.69, *p* = 0.000	F = 42.62, *p* = 0.000	F = 6.71, *p* = 0.000
**Sulfonamide**	10–500 mg/L	F = 514.31, *p* = 0.000	F = 49.23, *p* = 0.000	F = 11.39, *p* = 0.000
**Rifampicin**	5–250 mg/L	F = 8.08, *p* = 0.006	F = 73.18, *p* = 0.000	F = 3.39, *p* = 0.003
**Iron (III)/Phage > Ancestor**				
**Iron (III)**	100–1750 mg/L	F = 150.54, *p* = 0.000	F = 25.78, *p* = 0.000	F = 5.56, *p* = 0.000
**Iron (II)**	100–5000 mg/L	F = 1.63, *p* = 0.207	F = 0.24, *p* = 0.980	F = 4.80, *p* = 0.000
Gallium (III)	100–1750 mg/L	F = 854.93, *p* = 0.000	F = 69.20, *p* = 0.000	F = 14.33, *p* = 0.000
Silver (I)	5–100 mg/L	F = 62.23, *p* = 0.000	F = 10.41, *p* = 0.000	F = 13.99, *p* = 0.000
**Ampicillin**	20–500 mg/L	F = 374.13, *p* = 0.000	F = 7.71, *p* = 0.000	F = 1.75, *p* = 0.108
**Tetracycline**	5–100 mg/L	F = 87.07, *p* = 0.000	F = 9.98, *p* = 0.000	F = 7.27, *p* = 0.000
**Chloramphenicol**	5–250 mg/L	F = 40.02, *p* = 0.000	F = 42.51, *p* = 0.000	F = 6.66, *p* = 0.000
**Sulfonamide**	10–500 mg/L	F = 1215.90, *p* = 0.000	F = 62.81, *p* = 0.000	F = 14.15, *p* = 0.000
**Rifampicin**	5–250 mg/L	F = 2762.29, *p* = 0.000	F = 52.99, *p* = 0.000	F = 51.72, *p* = 0.000
**Iron (III)/Phage > Iron (III) resistant**				
**Iron (III)**	100–1750 mg/L	F = 52.29, *p* = <0.001	F = 82.59, *p* = <0.001	F = 1.25, *p* = 0.283
**Ampicillin**	20–500 mg/L	F = 37.37, *p* = <0.001	F = 3.31, *p* = 0.004	F = 0.635, *p* = 0.475
**Tetracycline**	5–100 mg/L	F = 28.23, *p* = <0.001	F = 24.51, *p* = <0.001	F = 1.17, *p* = 0.335
**Iron (III)/Phage = Iron (III) resistant**				
**Chloramphenicol**	5–250 mg/L	F = 1.57, *p* = 0.216	F = 49.27, *p* = 0.000	F = 0.33, *p* = 0.950
**Iron(III)/Phage > Phage**				
**Iron (III)**	100–1750 mg/L	F = 21.25, *p* = <0.001	F = 60.46, *p* = <0.001	F = 1.55, *p* = 0.154
**Iron (II)**	100–5000 mg/L	F = 19.44, *p* = 0.000	F = 5.31, *p* = 0.000	F = 5.29, *p* = 0.000
Gallium (III)	100–1750 mg/L	F = 743.54, *p* = <0.001	F = 29.70, *p* = <0.001	F = 14.25, *p* = <0.001
**Ampicillin**	20–500 mg/L	F = 120.12, *p* = <0.001	F = 11.48, *p* = <0.001	F = 3.36, *p* = 0.003
**Tetracycline**	5–100 mg/L	F = 24.43, *p* = <0.001	F = 13.34, *p* = <0.001	F = 1.87, *p* = 0.084
**Chloramphenicol**	5–250 mg/L	F = 25.65, *p* = 0.000	F = 32.29, *p* = 0.000	F = 4.17, *p* = 0.001
Silver (I)	5–100 mg/L	F = 74.49, *p* = <0.001	F = 14.11, *p* = <0.001	F = 10.83, *p* = <0.001
**Sulfonamide**	5–250 mg/L	F = 345.86, *p* = 0.000	F = 20.55, *p* = 0.000	F = 4.51, *p* = 0.000
**Iron (III)/Phage = Phage**				
**Rifampicin**	5–250 mg/L	F = 3.70, *p* = 0.060	F = 71.98, *p* = 0.000	F = 3.88, *p* = 0.000
**Sulfonamide**	10–500 mg/L	F = 0.41, *p* = 0.523	F = 8.56, *p* = 0.000	F = 0.48, *p* = 0.868
Silver (I)	5–100 mg/L	F = 0.73, *p* = 0.398	F = 43.62, *p* = <0.001	F = 0.37, *p* = 0.932
**Iron (III) > Iron (III)/Phage**				
Rifampicin	5–250 mg/L	F = 4.19, *p* = 0.046	F = 67.85, *p* = 0.000	F = 0.96, *p* = 0.479
**(b)**				
**Substance**	**Range Tested**	**Population**	**Concentration**	**Interaction**
**Iron (III) > Phage**				
Iron (II)	100–5000 mg/L	F = 38.18, *p* = <0.001	F = 2.74, *p* = 0.013	F = 2.99, *p* = 0.007
Gallium (III)	100–1750 mg/L	F = 696.66, *p* = <0.001	F = 30.41, *p* = <0.001	F = 17.44, *p* = <0.001
Silver (I)	5–100 mg/L	F = 207.07, *p* = <0.001	F = 59.90, *p* = <0.001	F = 47.65, *p* = <0.001
Ampicillin	20–500 mg/L	F = 12.67, *p* = <0.001	F = 16.48, *p* = <0.001	F = 2.82, *p* = 0.011
Sulfonamide	5–250 mg/L	F = 79.40, *p* = 0.000	F = 3.30, *p* = 0.00	F = 0.368, *p* = 0.933
Chloramphenicol	5–250 mg/L	F = 21.67, *p* = 0.000	F = 38.99, *p* = 0.000	F = 4.22, *p* = 0.001
** *P* ** **hage > Iron (III)**				
Iron (III)	100–1750 mg/L	F = 9.53, *p* = 0.003	F = 146.84, *p* = <0.001	F = 1.95, *p* = 0.065
**Iron (III) = Phage**				
Tetracycline	5–100 mg/L	F = 0.20, *p* = 0.661	F = 25.51, *p* = <0.001	F = 3.78, *p* = <0.001
Rifampicin	5–250 mg/L	F = 0.01, *p* = 0.908	F = 50.28, *p* = 0.000	F = 3.13, *p* = 0.006
**Iron (III) > Control**				
Iron (III)	100–1750 mg/L	F = 10.88, *p* = 0.002	F = 133.21, *p* = <0.001	F = 4.67, *p* = <0.001
Iron (II)	100–5000 mg/L	F = 40.58, *p* = <0.001	F = 1.84, *p* = 0.090	F = 3.34, *p* = 0.004
Gallium (III)	100–1750 mg/L	F = 653.99, *p* = <0.001	F = 27.88, *p* = <0.001	F = 15.72, *p* = <0.001
Tetracycline	5–100 mg/L	F = 60.13, *p* = <0.001	F = 33.82, *p* = <0.001	F = 21.11, *p* = <0.001
Ampicillin	20–500 mg/L	F = 82.44, *p* = <0.001	F = 9.62, *p* = <0.001	F = 1.31, *p* = 0.258
Chloramphenicol	5–250 mg/L	F = 43.62, *p* = 0.000	F = 65.78, *p* = 0.000	F = 7.53, *p* = 0.000
Sulfonamide	5–250 mg/L	F = 50.85, *p* = 0.000	F = 3.33, *p* = 0.004	F = 0.33, *p* = 0.950
Rifampicin	5–250 mg/L	F = 19.17, *p* = 0.000	F = 53.98, *p* = 0.000	F = 4.14, *p* = 0.001
**Iron (III)>Ancestor**				
Iron (III)	100–1750 mg/L	F = 131.19, *p* = <0.001	F = 65.68, *p* = <0.001	F = 11.74, *p* = <0.001
Silver (I)	5–100 mg/L	F = 155.21, *p* = <0.001	F = 42.20, *p* = <0.001	F = 53.63, *p* = <0.001
Gallium (III)	100–1750 mg/L	F = 756.28, *p* = <0.001	F = 60.10, *p* = <0.001	F = 19.12, *p* = <0.001
Ampicillin	20–500 mg/L	F = 151.14, *p* = <0.001	F = 12.71, *p* = <0.001	F = 0.76, *p* = 0.637
Tetracycline	5–100 mg/L	F = 123.49, *p* = <0.001	F = 59.99, *p* = <0.001	F = 40.17, *p* = <0.001
Chloramphenicol	5–250 mg/L	F = 58.84, *p* = 0.000	F = 32.16, *p* = 0.000	F = 25.73, *p* = 0.000
Sulfonamide	5–250 mg/L	F = 103.22, *p* = 0.000	F = 3.49, *p* = 0.003	F = 0.368, *p* = 0.933
Rifampicin	5–250 mg/L	F = 2206.29, *p* = 0.000	F = 25.69, *p* = 0.000	F = 26.32, *p* = 0.000
**(c)**				
**Substance**	**Range Tested**	**Population**	**Concentration**	**Interaction**
**Phage > Control**				
**Iron (III)**	100–1750 mg/L	F = 27.05, *p* = <0.001	F = 59.26, *p* = <0.001	F = 1.33, *p* = 0.242
**Ampicillin**	20–500 mg/L	F = 64.16, *p* = <0.001	F = 36.35, *p* = <0.001	F = 1.20, *p* = 0.317
**Tetracycline**	5–100 mg/L	F = 19.44, *p* = <0.001	F = 5.66, *p* = <0.001	F = 3.48, *p* = 0.003
**Rifampicin**	5–250 mg/L	F = 18.29, *p* = 0.000	F = 61.51, *p* = 0.000	F = 1.53, *p* = 0.169
**Sulfonamide**	5–250 mg/L	F = 20.90, *p* = 0.000	F = 6.99, *p* = 0.000	F = 0.59, *p* = 0.779
**Phage = Control**				
**Iron (II)**	100–5000 mg/L	F = 3.59, *p* = 0.064	F = 1.05, *p* = 0.414	F = 1.40, *p* = 0.220
Gallium (III)	100–1750 mg/L	F = 0.015, *p* = 0.903	F = 6.86, *p* = <0.001	F = 1.06, *p* = 0.407
Silver (I)	5–100 mg/L	F = 0.536, *p* = 0.467	F = 1.66, *p* = 0.129	F = 2.14, *p* = 0.047
**Chloramphenicol**	5–250 mg/L	F = 0.049, *p* = 0.826	F = 35.88, *p* = 0.000	F = 1.27, *p* = 0.279
**Phage > Ancestor**				
**Iron (III)**	100–1750 mg/L	F = 149.61, *p* = 0.000	F = 43.78, *p* = 0.000	F = 5.66, *p* = 0.000
**Tetracycline**	5–100 mg/L	F = 26.74, *p* = 0.000	F = 6.84, *p* = <0.000	F = 4.09, *p* = 0.001
**Ampicillin**	20–500 mg/L	F = 168.61, *p* = <0.000	F = 47.16, *p* = <0.000	F = 3.05, *p* = 0.007
**Rifampicin**	5–250 mg/L	F = 2198.97, *p* = 0.000	F = 36.08, *p* = 0.000	F = 27.82, *p* = 0.000
**Phage = Ancestor**				
**Chloramphenicol**	5–250 mg/L	F = 1.68, *p* = 0.200	F = 12.78, *p* = 0.000	F = 9.65, *p* = 0.000
**Sulfonamide**	5–250 mg/L	F = 0.31, *p* = 0.580	F = 7.36, *p* = 0.000	F = 1.46, *p* = 0.195
**Ancestor > Phage**				
**Iron (II)**	100–5000 mg/L	F = 11.76, *p* = 0.001	F = 1.32, *p* = 0.253	F = 1.46, *p* = 0.194
Gallium (III)	100–1750 mg/L	F = 11.53, *p* = 0.001	F = 31.11, *p* = 0.000	F = 11.35, *p* = 0.000
Silver (I)	5–100 mg/L	F = 4.91, *p* = 0.031	F = 2.52, *p* = 0.021	F = 9.51, *p* = 0.000
**(d)**				
**Substance**	**Range tested**	**Population**	**Concentration**	**Interaction**
**Control > Ancestor**				
Iron (III)	100–1750 mg/L	F = 42.01, *p* = <0.001	F = 30.36, *p* = <0.001	F = 2.26, *p* = 0.033
Gallium (III)	100–1750 mg/L	F = 9.40, *p* = 0.003	F = 27.20, *p* = <0.001	F = 7.53, *p* = <0.001
Ampicillin	20–500 mg/L	F = 12.16, *p* = <0.001	F = 25.87, *p* = <0.001	F = 1.39, *p* = 0.221
Chloramphenicol	5–250 mg/L	F = 7.55, *p* = 0.008	F = 58.18, *p* = 0.000	F = 38.75, *p* = 0.000
Rifampicin	5–250 mg/L	F = 1123.30, *p* = 0.000	F = 38.25, *p* = 0.000	F = 30.36, *p* = 0.000
Sulfonamide	5–250 mg/L	F = 83.30, *p* = 0.000	F = 17.64, *p* = 0.000	F = 1.26, *p* = 0.286
**Control = Ancestor**				
Silver (I)	5–100 mg/L	F = 0.94, *p* = 0.337	F = 2.39, *p* = 0.028	F = 3.24, *p* = 0.004
Tetracycline	5–100 mg/L	F = 0.39, *p* = 0.536	F = 3.68, *p* = 0.002	F = 0.28, *p* = 0.970
**Ancestor > Control**				
Iron (II)	100–5000 mg/L	F = 15.02, *p* = 0.000	F = 1.56, *p* = 0.160	F = 1.28, *p* = 0.273

**Table 2 antibiotics-12-00603-t002:** Ranking of populations based on fitness in excess iron (III) and bacteriophage T7. The ranking of each population is shown relative to excess iron (III) and bacteriophage T7. The fact that the phage-selected population performed better than the controls or ancestors indicates that selection for bacteriophage resistance has pleiotropic effects on excess iron resistance. Similarly, the fact that the iron (III)-resistant population performed better against bacteriophage than the controls or the ancestor shows that excess iron (III) resistance has pleiotropic effects on bacteriophage resistance.

Populations	Iron (III)	Phage T7
Iron (III)/Phage-resistant	1	1
Phage-resistant	3	2
Iron (III)-resistant	2	3
Control	4	4
Ancestor	5	5

**Table 3 antibiotics-12-00603-t003:** (**a**) Position of selective sweeps (yellow) and significant polymorphisms (green) in iron (III)-resistant populations at day 35. (**b**) Annotation of genes mutated (red-nucleotides changed) in iron (III)-resistant populations at day 35. Selective sweep is determined by an increase in frequency of a de-novo mutation to 1.000. A significant polymorphism is determined by an increase in frequency of a de-novo mutation to between 0.500 to 0.999. Stop codons are symbolized as *.

(a)											
Gene	Position	Mutation	Fe1	Fe2	Fe3	Fe4	Fe5	Fe6	Fe7	F8	F9/F10
*rpoC* →	4,185,540	C→T	0.078	0.000	0.000	1.000	0.000	0.000	0.000	0.000	0.000
*rpoB* →	4,183,399	Δ6 bp	0.000	0.000	0.000	0.000	0.000	0.000	0.935	0.000	1.000
*rpoB* →	4,183,204	G→T	0.000	0.000	0.000	0.000	0.000	0.902	0.000	0.063	0.000
*rpoB* →	4,181,278	C→T	0.000	0.000	0.000	0.000	0.801	0.000	0.000	0.000	0.000
*rpoC* →	4,187,633	A→C	0.000	0.713	0.000	0.000	0.000	0.000	0.000	0.000	0.000
*yeaG* →	1,868,570	G→T	0.000	0.647	0.000	0.000	0.000	0.000	0.000	0.000	0.000
*rpoB* →	4,184,809	G→A	0.000	0.000	0.000	0.000	0.000	0.000	0.527	0.000	0.000
*rpoB* →	4,184,795	C→G	0.000	0.000	0.526	0.000	0.000	0.000	0.000	0.000	0.000
*ycgB* ←	1,236,863	G→A	0.000	0.000	0.000	0.000	0.000	0.000	0.000	0.637	0.000
**(b)**											
**Gene**					**Annotation**						
*rpoC* →					P64L (CCG→CTG)						
*rpoB* →					D654Y (GAC→TAC)						
*rpoB* →					R12C (CGT→TGT)						
*rpoC* →					N762H (AAC→CAC)						
*yeaG* →					E555 * (GAG→TAG)						
*rpoB* →					G1189S (GGT→AGT)						
*rpoB* →					T1184R (ACG→AGG)						
*ychE* →/→ *oppA*					intergenic (+254/-485)						
*ycgB* ←					H127Y (CAT→TAT)						
*rpoB* →					coding (2155-2160/4029 nt)						

**Table 4 antibiotics-12-00603-t004:** (**a**) Position of selective sweeps (yellow) and significant polymorphisms (green) in phage-resistant populations at day 35. (**b**) Annotation of genes mutated (red-nucleotides changed) in phage-resistant populations at day 35. Selective sweep is determined by an increase in frequency of a de-novo mutation to 1.000. A significant polymorphism is determined by an increase in frequency of a de-novo mutation between 0.500 to 0.999. Stop codons are symbolized as *.

(a)						
Gene	Position	Mutation	Ph2	Ph3	Ph4	Ph5
*clpX* **→**/**→** *lon*	458,790	IS*186* (-) + 6 bp:: Δ1 bp	1.000	0.000	1.000	1.000
*clpX* →/→ *lon*	458,790	IS186 (+) +6 bp:: Δ1 bp	1.000	1.000	0.000	0.000
*uxaB* **←/←** *yneF*	1,610,807	T→C	0.000	0.000	0.000	1.000
*yeaG* **→**	1,868,147	G→A	0.000	1.000	0.000	0.000
*proQ* **←**	1,915,478	Δ1 bp	0.000	0.771	0.000	0.000
*fliR* **→**/**→** *rcsA*	2,023,824	Δ1 bp	1.000	0.000	0.000	0.000
*[gatR]*–*[fbaB]*	2,171,429	Δ6547 bp	0.000	0.000	0.000	1.000
*yejM* **→**	2,285,441	C→T	0.000	0.787	0.000	0.000
*ypjF* **→/←** *ypjA*	2,777,982	Δ1 bp	0.000	0.000	0.000	1.000
*rpoS* **←**	2,867,175	C→A	0.536	0.000	0.000	0.000
*rpoS* **←**	2,867,178	A→T	0.000	0.000	1.000	0.000
*rpoS* **←**	2,867,428	C→A	0.183	0.000	0.000	1.000
*rpoS* **←**	2,867,322	+CTT	0.000	0.868	0.000	0.000
*waaC* **→**	3,796,019	Δ1 bp	0.000	1.000	0.000	0.000
*waaC* **→**	3,796,167	IS*3* (+) +3 bp	0.000	0.000	1.000	0.000
*rpoC* **→**	4,186,532	A→G	0.000	0.000	1.000	0.000
*rpoC* **→**	4,187,522	A→C	0.827	0.000	0.000	0.000
*pgi* **→**/**→** *yjbE*	4,235,682	T→C	0.000	0.000	0.000	1.000
*yjbH* **→**	4,237,938	A→C	0.000	0.000	0.000	1.000
*yjbH* **→**	4,238,073	G→A	0.000	0.000	1.000	0.000
*yjbH* **→**	4,239,443	Δ5 bp	0.704	0.000	0.000	0.000
**(b)**						
**Gene**		**Annotation**				
*clpX* **→**/**→** *lon*		intergenic (+90/−93)				
*clpX* →/→ *lon*		intergenic (+90/−93)				
*uxaB* **←/←** *yneF*		intergenic (−127/+100)				
*yeaG* **→**		E414K (GAA→AAA)				
*proQ* **←**		coding (57/699 nt)				
*fliR* **→**/**→** *rcsA*		intergenic (+146/−144)				
*[gatR]*–*[fbaB]*		IS*3*-mediated				
*yejM* **→**		Q356 * (CAG→TAG)				
*ypjF* **→/←** *ypjA*		intergenic (+200/+164)				
*rpoS* **←**		G126V (GGG→GTG)				
*rpoS* **←**		L125Q (CTG→CAG)				
*rpoS* **←**		E42 * (GAA→TAA)				
*rpoS* **←**		coding (230/993 nt)				
*waaC* **→**		coding (41/960 nt)				
*waaC* **→**		coding (189–191/960 nt)				
*rpoC* **→**		K395E (AAA→GAA)				
*rpoC* **→**		M725L (ATG→CTG)				
*pgi* **→**/**→** *yjbE*		intergenic (+275/−224)				
*yjbH* **→**		Y102S (TAT→TCT)				

**Table 5 antibiotics-12-00603-t005:** (**a**) Position of selective sweeps (yellow) and significant polymorphisms (green) in iron (III)/phage-resistant populations at day 35. (**b**) Annotation of genes mutated (red-nucleotides changed) in iron (III)/phage-resistant populations at day 35. Selective sweep is determined by an increase in frequency of a de-novo mutation to 1.000. A significant polymorphism is determined by an increase in frequency of a de-novo mutation between 0.500 to 0.999.

(a)							
Gene	Position	Mutation	FPh1	FPh2	FPh4	FPh5	FPh6
*rcsA →*	2,024,505	A→T	0.867	1.000	0.842	1.000	0.721
*hldE ←*	3,195,969	G→T	0.000	0.000	0.000	1.000	0.000
*waaC →*	3,796,783	IS1 (+) +9 bp	1.000	0.000	0.000	0.000	0.000
*rpoB →*	4,183,399	Δ6 bp	0.000	0.000	0.474	1.000	0.521
*nudE ←*/*→ yrfF*	3,526,449	IS5 (+) +4 bp	0.000	0.000	0.000	0.628	0.000
*hldD →*	3,794,149	+GA:: IS3 (+) +4 bp	0.000	0.895	0.159	0.000	0.238
*rpoC →*	4,187,507	A→C	0.726	0.000	0.000	0.000	0.000
**(b)**							
**Gene**		**Annotation**					
*rcsA* *→*		I180F (ATC→TTC)					
*hldE* *←*		A262E (GCG→GAG)					
*waaC* *→*		coding (805–813/960 nt)					
*rpoB* *→*		coding (2155–2160/4029 nt)					
*nudE ←*/*→ yrfF*		intergenic (−300/−17)					
*hldD* *→*		coding (163–166/933 nt)					
*rpoC* *→*		N720H (AAC→CAC)					

## Data Availability

The SRA accession number for sequencing data from this study is PRJNA803149 for Iron (III)-, Phage-, Iron (III)/Phage-resistant, and control populations.

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
