# Peer review of "Serving Two Masters: Effect of *Escherichia coli* Dual Resistance on Antibiotic Susceptibility"

_antibiotics, 2023, doi:10.3390/antibiotics12030603_

Round 1
Reviewer 1 Report
I found this article interesting for the readers and followed the journal Antibiotics’ scope. I don’t have any major comments as this article has enough data, however, the author could have discussed SAR of benzamidine analogs to make this article more interesting to the reader of Antibiotics.
I would recommend the article be published in Antibiotics after minor corrections.
The author needs to address the following comments/corrections.
1. All mg/ml or μl need to change mg/mL or μL.
2. All footnotes’ fonts should be different than the font of texts.
3. The author could have mentioned the mechanism of resistance to iron (III).
4. Change “-20⁰ C” to “-20 oC”, and all similar errors.
5. The author should correct the format of references wherever needed (e.g Year Bold, Volume Italic etc).
6. The author could include the following relevant references.
(a) Tyerman JG, Ponciano JM, Joyce P, Forney LJ, Harmon LJ. The evolution of antibiotic susceptibility and resistance during the formation of Escherichia coli biofilms in the absence of antibiotics. BMC Evol Biol. 2013 Jan 28;13:22. doi: 10.1186/1471-2148-13-22. PMID: 23356665; PMCID: PMC3568021.
(b) Gu H, Lee SW, Carnicelli J, Jiang Z, Ren D. Antibiotic Susceptibility of Escherichia coli Cells during Early-Stage Biofilm Formation. J Bacteriol. 2019 Aug 22;201(18):e00034-19. doi: 10.1128/JB.00034-19. PMID: 31061169; PMCID: PMC6707912.
Reviewer 2 Report
Nice work with important results and conclusions.
Minor suggestions:
59-60 lines need a bibliography reference
115 line: why were these drugs chosen-needs expanation
Reviewer 3 Report
Review of Antibiotics-2237554
In this paper, the authors use experimental evolution of E. coli to develop resistance to excess iron and/or phage T7 and examine the growth of these evolved bacteria to iron and other metals, various antibiotics and phage T7.
This is potentially interesting, as evolution of resistance to one stressor often leaves bacteria with lower fitness and greater susceptibility to other stressors. In fact, in this study, selection for iron resistance did not affect the evolution of phage resistance, nor did iron or phage resistance cause greater sensitivity to antibiotics. Indeed, iron resistance appears to have conferred resistance to antibiotics. What is the mechanism for development of antibiotic resistance as a result of selection on excess iron? Is it related to upregulation of an efflux pump?
Unfortunately, the paper is somewhat confusingly written, dense with numbers and a tedious read, but short on explanation. Several pieces of data are not described in the results, and tables are not referenced. The figure quality is low and not clearly explained. Overall, it seems a bit sloppy and hastily put together.
Although there is sequence data for the evolved bacteria in Tables 4, 5 and 6, these tables are not referred to in the text, and a description of these results appears to be missing from the results section. The tables are impossible to understand without any legends. The mutation data is mentioned in the discussion, however, and some of the mutations recovered are potentially interesting. In some cases, they are relatively easy to explain, for example those related to LPS synthesis for phage resistance. In many cases, however, their relevance to the resistance mechanisms is hard to judge without further validation.
There is a conclusion section after the materials and methods. This should be moved to immediately after the discussion. In fact, I think the journal format calls for Materials and Methods coming immediately after the Introduction. It would also be helpful if the conclusion of the study could be stated at the end of the introduction.
Lines 78-80:
The results section starts with a description of growth of resistant vs non-resistant populations; however, the creation of these resistant populations has not yet been described at this point. It seems to me that a section describing the experimental evolution of these populations needs to come first. What is the “control”? Presumably it is bacteria that have been evolved in the absence of iron and phage. This needs to be more clearly described.
Figure 1d indicates that growth is better at high silver concentration than low. Why?
Figure 3: confusing use of small letter both to indicate the main segments of the figure (a, b) as well as the sub-panels (a-e) in part panel b. Use one unified labeling system (e.g. a-f). In fact, the pictures of the plates in panel b seem superfluous, as the values are already represented in the bar graph. Maybe they could be moved to supplementary data.
Figure 4: Same comment as for Figure 3: labeling is confusing and panel (b) is superfluous. The legend to (a) does not explain what the bar graph shows. (This is stated in the text, but the figure legends should be self-explanatory.)
Figure 5 is missing labels. There is no explanation of what we are actually looking at.
Line 53: no comma after Bacteria
Line 78: showed shows
Line 434: What is “0.86% saline”? 0.86% (w/v) NaCl in water?
Round 2
Reviewer 3 Report
The revised version of the manuscript has been improved, and there is now a section (3.4) describing the genomics data. I also appreciate the addition of an explanation of the general linear model in the legend to Table 1a.
The presentation still leaves a bit to be desired and much of the data is presented without much explanation to help the non-expert. It’s pretty difficult to grasp what all the F-values shown in Table 1a-d represent. Is there a way to extract the main points out of all these data and represent it in a more obvious and visual way?
Likewise, there is no explanation of the numbers in Tables 3 and 4. It is not immediately obvious to the non-expert why these numbers represent selective sweeps or “significant polymorphisms.” (Presumably they represent the fraction of reads that contained the mutation—this should be stated)
In the text, r(rho)-value is used, but in the table it is P (and in the legend it is p). Please be consistent.
Line 70: The authors misunderstood my comment. There should be no comma after "bacteria" on line 70.
174: 0.86% saline should be 0.86% (w/v) NaCl
Figure 5 still has no explanation. Presumably a vertical line of phage was dispensed onto the plate, but this is not mentioned here or in the methods.
Line 402: “selected for T7 phage”. Did they mean selected for resistance to T7 phage?
